# Deciphering conformational selectivity in the A$_{2A}$ adenosine G protein-coupled receptor by free energy simulations

**Willem Jespers**[1,2]*, **Laura H. Heitman**[2,3], **Adriaan P. IJzerman**[2], **Eddy Sotelo**[4,5], **Gerard J. P. van Westen**[2], **Johan Åqvist**[1], **Hugo Gutiérrez-de-Terán**[1,6]*

**1** Department of Cell and Molecular Biology, Uppsala University, Biomedical Center (BMC), Uppsala, Sweden, **2** Drug Discovery and Safety, Leiden Academic Centre for Drug Research, Leiden, The Netherlands, **3** Oncode Institute, Leiden, Leiden, **4** Centro Singular de Investigación en Química Biolóxica y Materiais Moleculares (CIQUS), Santiago de Compostela, Spain, **5** Departamento de Química Orgánica, Facultade de Farmacia, Universidade de Santiago de Compostela, Santiago de Compostela, Spain, **6** Science for Life Laboratories, BMC, Uppsala, Sweden

* w.jespers@lacdr.leidenuniv.nl (WJ); hugo.gutierrez@icm.uu.se (HGT)

## Abstract

Transmembranal G Protein-Coupled Receptors (GPCRs) transduce extracellular chemical signals to the cell, via conformational change from a resting (inactive) to an active (canonically bound to a G-protein) conformation. Receptor activation is normally modulated by extracellular ligand binding, but mutations in the receptor can also shift this equilibrium by stabilizing different conformational states. In this work, we built structure-energetic relationships of receptor activation based on original thermodynamic cycles that represent the conformational equilibrium of the prototypical A$_{2A}$ adenosine receptor (AR). These cycles were solved with efficient free energy perturbation (FEP) protocols, allowing to distinguish the pharmacological profile of different series of A$_{2A}$AR agonists with different efficacies. The modulatory effects of point mutations on the basal activity of the receptor or on ligand efficacies could also be detected. This methodology can guide GPCR ligand design with tailored pharmacological properties, or allow the identification of mutations that modulate receptor activation with potential clinical implications.

## Author summary

The design of new ligands as chemical modulators of G protein-coupled receptors (GPCRs) has benefited considerably during the last years of advances in both the structural and computational biology disciplines. Within the last area, the use of free energy calculation methods has arisen as a computational tool to predict ligand affinities to explain structure-affinity relationships and guide lead optimization campaigns. However, our comprehension of the structural determinants of ligands with different pharmacological profile is scarce, and knowledge of the chemical modifications associated with an agonistic or antagonistic profile would be extremely valuable. We herein report an original implementation of the thermodynamic cycles associated with free energy perturbation

**Data Availability Statement:** All data necessary to reproduce the simulations, together with the output used to generate all figures and tables, is

accessible at https://zenodo.org/record/5602896#.
YXlGHZ5ByF4. The code used to perform and
analyze the simulations and produce the figures
and tables is freely available at https://github.com/
qusers/qligfep and https://github.com/esguerra/q6.

**Funding:** JÅ is co-recipient of a Knutt and Alice
Wallemberg Fundation (Evolution of new genes
and proteins, https://kaw.wallenberg.org/en/
research/focus-fundamental-evolutionary-biology)
ES received finantial support from Consellería de
Cultura, Educación e Ordenación Universitaria
[Galician Government: (grant: ED431B 2020/43),
the Xunta de Galicia (Centro singular de
investigación de Galicia accreditation 2019-2022,
ED431G 2019/03). http://www.edu.xunta.gal The
funders had no role in study design, data collection
and analysis, decision to publish, or preparation of
the manuscript.

**Competing interests:** The authors have declared
that no competing interests exist.

(FEP) simulations, to mimic the conformational equilibrium between active and inactive
GPCRs, and establish a framework to describe pharmacological profiles as a function of
the ligands selectivity for a given receptor conformation. The advantage of this method
resides into its simplicity of use, and the only consideration of active and inactive confor-
mations of the receptor, with no simulation of the transitions between them. This model
can accurately predict the pharmacological profile of series of full and partial agonists as
opposed to antagonists of the $A_{2A}$ adenosine receptor, and moreover, how certain muta-
tions associated with modulation of basal activity can influence this pharmacological pro-
files, which enables our understanding of such clinically relevant mutations.

## Introduction

G Protein-Coupled Receptors (GPCRs) are membrane proteins that transduce the signals of
hormones, neurotransmitters and metabolites into an appropriate cellular response [1]. The
canonical intracellular signalling pathways are mediated by heterotrimeric G proteins, though
alternative pathways exist like those involving β-arrestin proteins. GPCRs are widely involved
in human physiology, where over 800 genes encode six GPCR classes [2] and they constitute
the main target of approximately 34% of marketed drugs [3]. Our knowledge of the structure-
function relationships of GPCRs has increased tremendously in the last decades, largely fuelled
by a growing number of GPCR structures. The first crystal structures corresponded to inactive
states, following strategies such as fusing the receptor with stabilizing proteins [4] or the intro-
duction of state-specific thermostabilizing mutations [5]. The last approach allowed obtaining
the first structures in a pseudo-active state in complex with an agonist, where the receptor
showed the initial conformational changes characteristic of activation but still lacking any
intracellular binding partner [6]. The structures of ternary GPCR complexes, including the
intracellular signalling G proteins or β-arrestin in addition to an orthosteric agonist, were
resolved in recent years mainly due to advances in cryo-EM [7,8]. Traditionally, GPCR activa-
tion has been described as a two-state model, where the receptor would transition from an
inactive (R) to an active (R*) conformation (Fig 1A) [9]. This relatively simplistic model
becomes more realistic when considering the influence on the receptor equilibrium of chemi-
cal modulators (Fig 1B), receptor mutations (Fig 1C) or even the intracellular signalling pro-
tein [10].

The four adenosine receptors (ARs), namely $A_1$, $A_{2A}$, $A_{2B}$ and $A_3$, constitute one of the best
structurally characterized GPCR families [11]. With more than 50 entries in the PDB, the
$A_{2A}AR$ was one of the first receptors to be captured in the three different conformational states
(inactive, active-like, ternary complex) [12–14], and was later accompanied by the inactive and
ternary complexes of the $A_1AR$ [15,16]. Structural and mutational data of $A_{2A}AR$ revealed that
conformational changes associated with activation involve a widening of the 'ribose pocket',
lined by residues $T88^{3.36}$, $S277^{7.42}$ and $H278^{7.43}$ [upper case numbers refer to Ballesteros-Wein-
stein numbering] [17,18]. All adenosine receptor full agonists known to date contain a ribose
moiety (Fig 2), with the hydroxy substituents forming hydrogen bonds with these residues. As
such, the stereospecificity of the ribose group is important in receptor activation. On the other
hand, partial agonists are molecules that display a reduced maximum efficacy as compared to
the full agonists. In the case of $A_{2A}AR$, partial agonists have been reported bearing as most as
one hydroxy substituent, which is supposed to form a single hydrogen bond interaction with
one of these residues in the ribose pocket (Fig 2). Even compounds with no hydroxyl can
behave as partial agonists (i.e. LUF5833, Fig 2), in which case the mechanistic hypothesis for

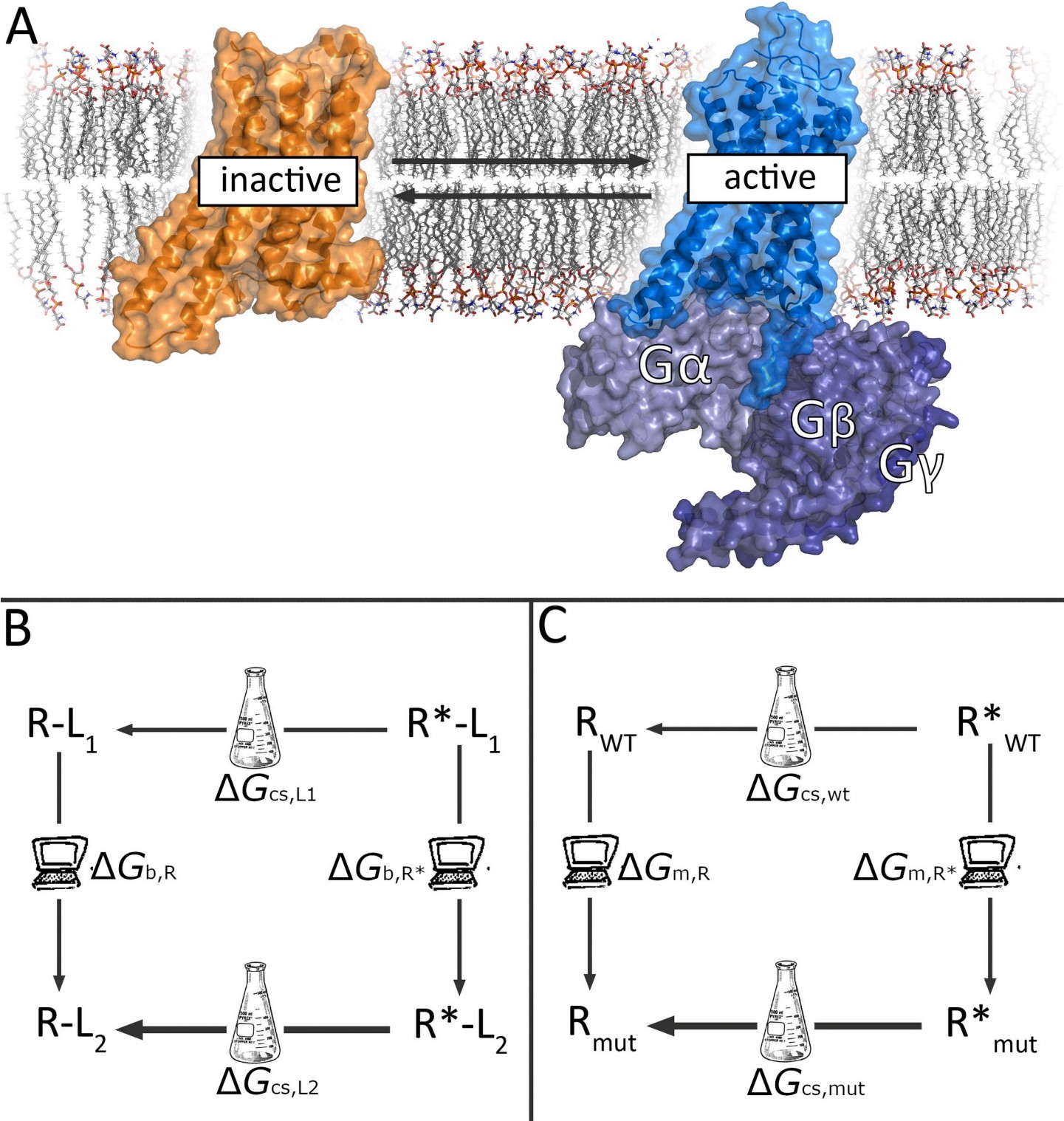

**Fig 1. Thermodynamic cycles and receptor activation states.** (A) Two-state model of GPCR activation between the inactive (orange) and active structure bound to a G protein (blue). (B) Thermodynamic cycle linking ligand efficacy to relative affinities for a given receptor conformation. The experimental (horizontal) legs represent the effect of a neutral antagonist (L2) on the receptor equilibrium (with no expected conformational selection, CS), and the induced shift towards the active state (R*) due to agonist (L1) binding (represented by a thinner line towards the inactive conformation). (C) the receptor basal equilibrium can be shifted by a certain point mutation (mut), in this example a CIM that selectively stabilizes the inactive state (thicker line). Each thermodynamic cycle can be closed with vertical legs, representing the corresponding FEP calculations between the ligand pair (B) or the receptor mutation (C), allowing to estimate the differences in experimental values as the calculated difference in the FEP legs.

**Fig 2. Chemical structures of the compounds considered in this work, classified according to their experimental pharmacological profile.**

its partial efficacy could rely on an optimal steric fitting of the phenyl substituent in the enlarged ribose pocket of the active state.

Site-directed mutagenesis has been traditionally used in GPCR research to characterize the shifts in ligand binding affinities induced by point mutations, and used to assist the elucidation of ligand binding modes as recently reviewed for the AR family [18]. Two positions invariably affect both agonist and antagonist ligand binding, namely N253$^{6.55}$ and F168$^{EL2}$, which were later confirmed as anchoring points for all heterocycles present in currently described orthosteric ligands [18]. Point mutations can also influence the thermal stability of the receptor, either its overall stability or selectively increasing the stability of one receptor state or conformation (Fig 1C). Moreover, one mutation can simultaneously affect ligand affinity whilst also shifting the receptor conformational equilibrium. For instance, mutations S277$^{7.42}$A and T88$^{3.36}$A, positioned in the ribose pocket, increase the affinity of antagonists while the efficacy of agonists is reduced. In addition to this effect, S277$^{7.42}$A increases the efficacy of some partial agonists [19], while the T88$^{3.36}$A mutation effectively shifts the receptor equilibrium to the inactive state, being classified as a constitutively inactive mutation (CIM) since the basal activity is concomitantly decreased [20]. Thus, the overall effect on ligand binding affinity can be a consequence not only of direct interactions with the ligand, but also of the increased stability (and thus availability) of the receptor conformation that preferably binds a pharmacological class of ligand.

Based on the available collection of experimental structures, reliable models of AR-ligand complexes can be generated via computational methods [21,22]. While docking algorithms can be very useful for this goal, they typically fail to describe free energies of binding correctly [23]. Instead, the increased availability of computational power and algorithms have enabled a routine use of first-principle methods such as free energy perturbation (FEP) to accurately estimate ligand-binding free energies, also for GPCRs [24]. In this scenario, we have recently developed robust FEP protocols that were thoroughly applied in the context of GPCR ligand-binding investigations: QligFEP [25] allows to systematically compute relative binding affinity changes between a series of ligands, while QresFEP [26] was designed to evaluate the binding affinity shifts due to single point mutations. The synergistic combination of both approaches recently led to the elucidation of the binding mode of a series of A$_{2A}$AR antagonists [27]. In this work, we extend the applicability of these methods to study the effects of chemical modifications of ligands, as well as receptor mutations, on the conformational equilibrium of the receptor, which we relate to efficiency of the ligand as an agonist modulator. First, we analyse the structure-efficacy determinants of series of full and partial agonists. Thereafter, we elucidate the role of the S277$^{7.42}$A and T88$^{3.36}$A mutations in the associated conformational equilibrium of the A$_{2A}$AR receptor, to finally determine the specific role of these mutations on the efficacy of selected partial and full agonists. The outcome of this study can not only aid the design of chemical modulators with tailored pharmacological properties, but also be broadly applicable to characterize GPCR mutations with clinically relevant effects.

## Results

### Conformational selectivity of ligands depends on their pharmacological profile

In this first section, we explore the predicted conformational selectivity for a number of ligands, as a function of their pharmacological profiles. To do this, a thermodynamic cycle was designed to estimate, for a given molecular pair of e.g. agonist and antagonist, the relative affinities between the two relevant conformational states of the A$_{2A}$AR. The cycle is solved by subtracting the corresponding binding free energies ($\Delta G_b$), calculated via an FEP

transformation (agonist → antagonist) performed in the inactive state ($\Delta G_{b,R}$), from the same transformation performed in the active state ($\Delta G_{b,R^*}$, vertical legs in Fig 1B). The difference between these estimated affinities would correspond to the difference in the conformational selectivity between these two ligands ($\Delta\Delta G_{cs(L1-L2)}$) as:

$$\Delta\Delta G_{cs(L1-L2)} = \Delta G_{cs,L1} - \Delta G_{cs,L2} = \Delta G_{b,R*} - \Delta G_{b,R}$$

Such model assumes that the pharmacological profile of a ligand (i.e. agonist, antagonist) is dictated by a more or less biased selectivity towards the active conformation, respectively. In each case, the pair of agonist and antagonist molecules to compare should share enough chemical similarity (e.g. bearing the same chemical scaffold) while maintaining a sufficiently distinct pharmacological profile. This scheme was consequently used to investigate the variability in the pharmacological profile within three chemical scaffolds: *i)* the classical ribose-containing agonists, such as NECA; *ii)* 2-phenylaminothiazolo[5,4-d]pyrimidines, which were recently characterized as partial agonists depending on the substituent in position 7 [28]; and *iii)* partial agonists derived from the 4-phenylpyridine scaffold of LUF5834 [29]. In each case, the (partial/full) agonist was compared to a chemical analog that behaves as a (neutral) antagonist (Fig 2).

The experimentally determined coordinates of classical agonists binding to the $A_{2A}AR$ (Fig 3A and 3B) were used as a starting configuration for the active-state simulations (denoted with an asterisk), while the corresponding binding mode to the inactive $A_{2A}AR$ was generated by receptor superposition. The binding mode of 9-cyclopentyadenine (Cyp-Ade) antagonist where the ribose group of adenosine is replaced by a cyclopentane [30], was generated using a flexible ligand alignment strategy (see Methods). The initial configuration of the two partial agonist chemotypes was generated via docking to the $A_{2A}AR^*$, showing occupancy of the ribose pocket (Fig 3C and 3D), while the corresponding antagonist analogues were modeled in an analogous binding orientation in the $A_{2A}AR$ inactive structure. All compounds shared similar interactions with N253$^{6.55}$ and F168$^{EL2}$, irrespective of the receptor conformation or ligand chemotype.

The full agonists NECA and adenosine (ADO) show different experimental efficacies for the activation of $A_{2A}AR$. Consequently, NECA was set as a reference with a maximum efficacy of 100%, with adenosine having 45% efficacy relative to NECA [29], and the neutral antagonist Cyp-Ade having 0% efficacy. A qualitative descriptor, Δ-efficacy, can be defined as the positive difference in % efficacy values between pairs of (partial) agonist and antagonist compounds. In analogy, the thermodynamic cycle depicted in Fig 1B allows to estimate the relative preference for the active conformation for the same pair of molecules, a property that we will try to correlate with the corresponding efficacy shifts. Here, one has to note that, due to their fundamentally different formulation, a full quantitative correlation between the experimental (Δ-efficacy, percentage) and the calculated (ΔΔG, logarithmic) descriptors is not expected. However, a qualitative correlation would indicate that our end-state modeling of ligand efficacy would be useful to explain efficacy shifts between molecule pairs, and thus be of potential interest to further predict ligand pharmacological profiles. In other words, the differences in the calculated ΔΔG might be used to classify the compounds on the basis of their expected efficacy gain from a reference molecule. According to the scheme in Fig 1B, an expected positive value in the calculated ΔΔG would indicate the expected preference of the agonist for the active state. This was indeed the case for the NECA and ADO perturbations to the antagonist Cyp-Ade (Fig 4 and S1 Table). Moreover, the magnitude of the calculated ΔΔG value is double for the NECA transformation as compared to the case of adenosine, which indicates the correct ranking of this class of compounds according to their expected efficacy.

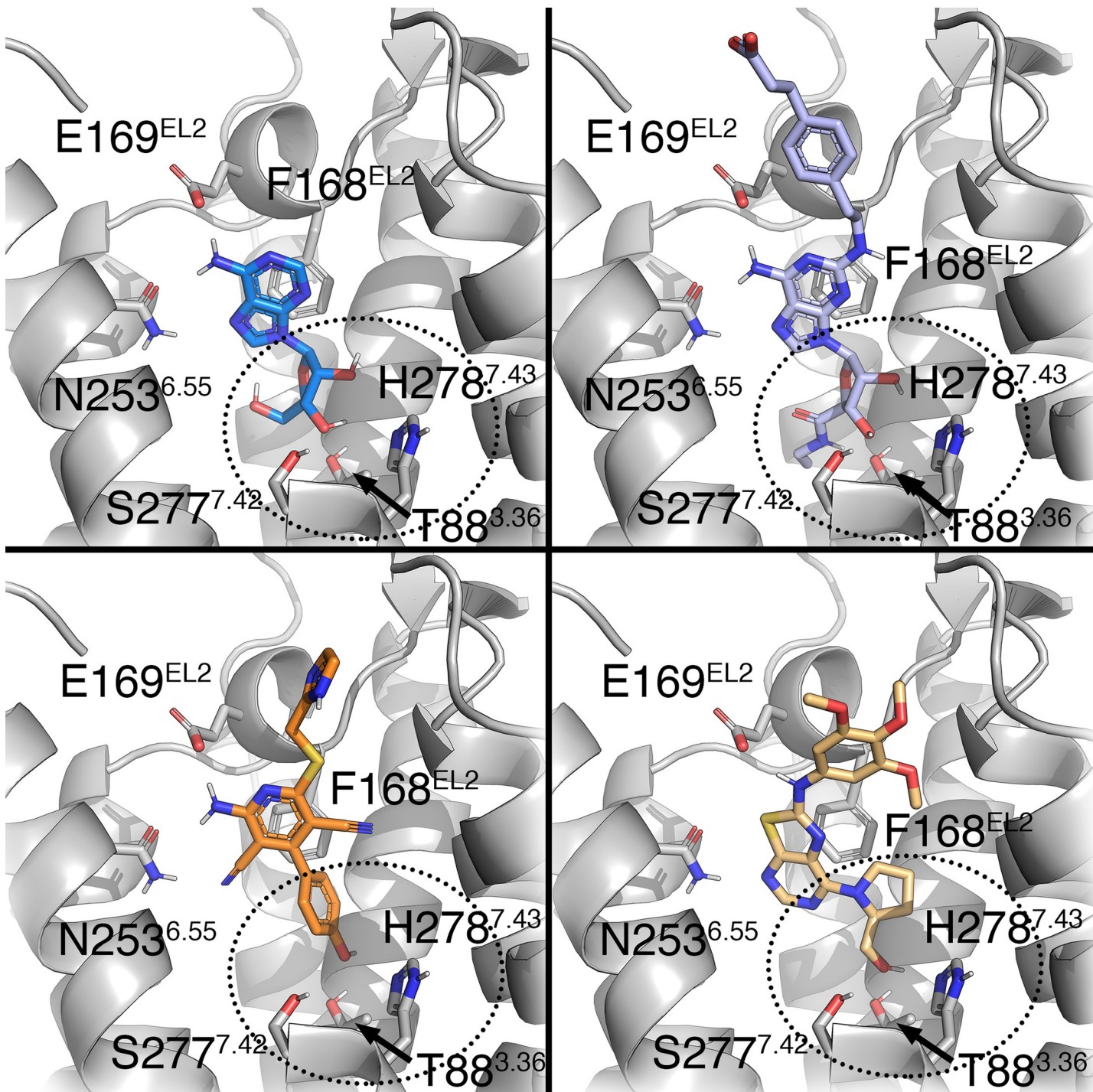

**Fig 3. Binding mode of four adenosine A$_{2A}$AR agonists with different efficacies.** (A, B) Top panels show the experimental pose of full agonists adenosine (A, PDB code 2YDV [31]) and CGS21680 (B, PDB code 4UG2 [13]). (C, D) Bottom panels show the docking model obtained for partial agonists LUF5834 (C) [29] and 10n (D) [28]. The key residues common for ligand-receptor interactions (N253$^{6.55}$, E169$^{EL2}$ and F168$^{EL2}$) [18] are shown in sticks. The ribose binding site is denoted in a dotted circle, with interacting residues associated with ligand activation shown in sticks.

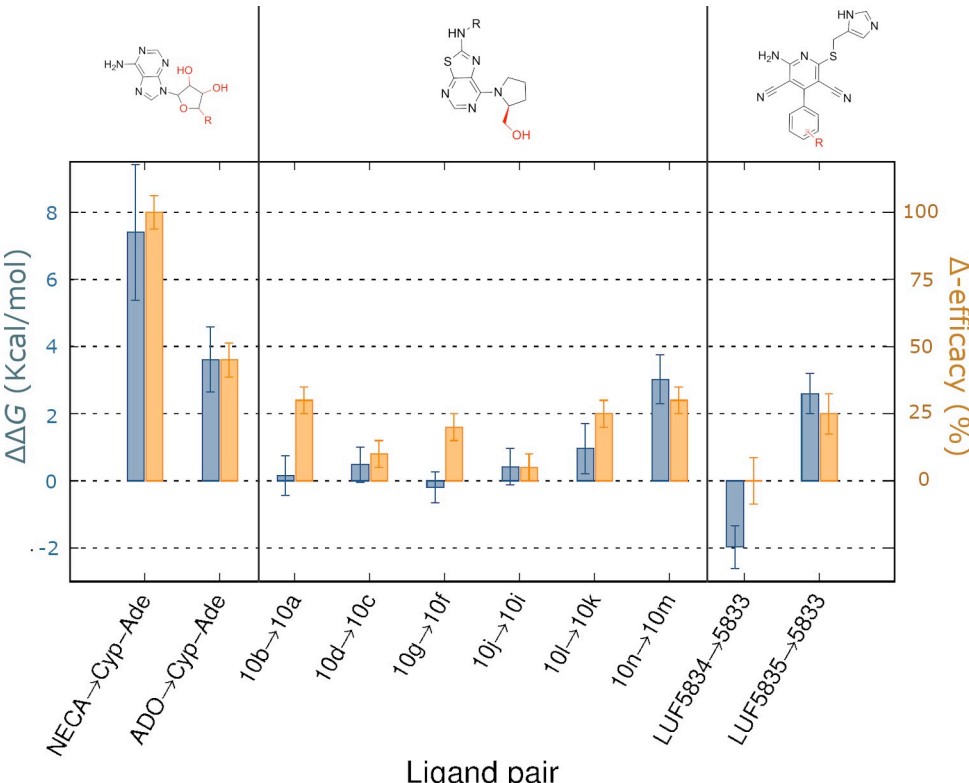

**Fig 4. Calculated relative affinities for the active *vs* inactive receptor conformation.** Relative affinities expressed as
$\Delta\Delta G = \Delta G_{b,R^*} - \Delta G_{b,R}$ (kcal/mol, blue bars, average values ± SEM) for pairs of agonists/antagonist of analogous
chemotype (depicted in the top, with the chemical groups varied along the FEP simulations in red). For each ligand
pair, the relative shift in experimental efficacy is shown as $\Delta$-efficacy (orange bars, $E_{max,ago}$-$E_{max,antago}$, in % ± SEM),
with corresponding Emax values reported relative to the reference full agonists NECA [28], or CGS21680 -for the LUF
chemotype [29], see text.

We then moved on to examine the molecular determinants of the agonistic properties for a
series of 7-(prolinol-N-yl)-2-phenylaminothiazolo[5,4-d]pyrimidines on the $A_{2A}AR$ [28]. The
SAR and earlier molecular modeling of these series revealed that the prolinol substituent was
essential to maintain the partial agonist profile, presumably by partially mimicking the ribose
interactions (see compounds, **10b, 10d, 10g, 10j, 10l, 10n,** Fig 2), in contrast to the corre-
sponding pyrrolidine substituted compounds (**10a, 10c, 10f, 10i, 10k and 10m**, Fig 2) which
are all neutral antagonists (i.e. with efficacy not significantly different from 0%) [28]. Following
our preliminary results on this chemotype [32], we systematically applied the thermodynamic
cycle in Fig 1B to all corresponding partial agonist/antagonist pairs within this series, the
results shown in Fig 4 and S1 Table. A deeper look into the experimental data shows that the
efficacy of the partial agonists is modulated by decorations of the exocyclic amino group (R,
see Fig 2). Our calculations indicate that compound **10m** would show the highest preference
for the active state of the receptor, followed by compound **10k** (Fig 4), in line with their efficacy
rank experimentally observed. Our modeling shows a tendency on prolinol-containing com-
pounds to favor the binding to the active receptor, as compared to their pyrrolidine substituted
compounds, with the exception of the pairs and **10g/f** and, to some extent, **10b/a** where no sig-
nificant conformational selection is calculated (Fig 4). Explaining the role of this substituent in
the partial agonist profile of this chemotype. The structural interpretation of this data rein-
forces the role of the hydroxyl group in the 7-prolinol in engaging the residues that trigger

$A_{2A}AR$ activation (Fig 3), explaining the importance of this substituent for a partial agonist profile.

The third chemotype here explored is derived from the 4-phenylpyridine scaffold, characteristic of a family of $A_{2A}AR$ partial agonists represented by LUF5833 (Fig 2) [29]. Hydroxy decorations on the phenyl ring have been shown to affect ligand efficacy: the *p*-OH-phenyl in LUF5834 is equipotent to the unsubstituted LUF5833 (55% ± 15% as compared to the efficacy of the reference full agonist CGS21680), while the *m*-OH-phenyl in LUF3835 results in an increased value of this relative efficacy of 80% ± 10% (see Fig 2) [19]. In their original report, Lane et al. postulated that these compounds bind with the phenyl substituent deep in the binding pocket of the $A_{2A}AR$, making different interactions with activation-related residues as compared to ribosidic agonists [19]. We herein examined if such a binding mode hypothesis (Fig 3) could explain the differences in ligand efficacy following the FEP approach outlined in Fig 1B. In this case, the dehydroxylated partial agonist LUF5833 was used as the reference ligand in two pair comparisons. Since the three compounds show comparable experimental binding *affinities* for the $A_{2A}AR$, the experimental increase in *efficacy* for a given derivative can be directly correlated with its increased relative affinity for the active ($R^*$) over inactive (R) receptor conformation, which is precisely the output of our calculations. The results (Fig 4 and S1 Table) indicate that the introduction of a *m*-OH-phenyl in LUF5835 would lead to a significant conformational selection for the active receptor, in line with the experimental gain of approximately 30% efficacy [19]. The structural interpretation of this predicted efficacy increase is located on the hydrogen bond between the OH group in *meta* position of LUF5835 with T88$^{3.36}$, observed in the simulations with $R^*$, an interaction that is well known to be involved in agonist recognition [18]. However, the introduction of a *p*-OH-phenyl substitution in the reference ligand leads to a reduced predicted efficacy, since this substituent would not be making any preferred interaction in the active state as compared to the inactive, with the experimental data showing equipotency of this compound pair.

During the preparation of this manuscript, a new crystal structure of compound LUF5833 in complex with the inactive conformation of the $A_{2A}AR$ was published [33]. Comparison of our model with this structure revealed an almost identical inactive conformation for the receptor ($RMSD_{A2AAR}$ = 0.78 Å for the Cα trace). The ligand position was also in good agreement with our MD simulations with an overall $RMSD_{LUF5833}$ = 2.20 ± 0.45 Å, calculated over the last 10% of the FEP trajectories. Specifically, most variability was located on the flexible 1H-imidazol- 2-ylmethylsulfanyl substituent oriented towards the extracellular cavity, with the core moiety being more stable along the MD simulation ($RMSD_{LUF5833\_core}$ = 1.38 ± 0.51 Å). The structure also demonstrates that the partial agonist LUF5833 can actually bind to the inactive conformation of the receptor, which supports the line of reasoning of this study.

## The effect of point mutations on basal activity and conformational selectivity

According to the experimental mutagenesis data, the activation trigger induced by the partial agonist LUF5834 would involve different interactions with residues in the ribose binding site, as compared to ribose-containing full agonists [19]. In that study, CGS21680, a C2-substituted variation of NECA (Figs 2 and 3), was used as a reference full agonist in the pharmacological characterization of the LUF series of compounds. Particularly intriguing was the effect of two mutations, T88$^{3.36}$A and S277$^{7.42}$A, on the modulation of the efficacy of these two molecules. While the binding affinity and potency of CGS21680 was severely reduced by both mutations, the potency of LUF5834 was unaltered or even slightly increased [19].

Consequently, we wondered about the molecular mechanism of the different mutational-induced shifts on the internal efficacy of the two ligands. This is indeed a complex question, as

one can imagine two mechanisms via which a mutation can modulate the efficacy of an agonist: On the one hand, the mutation might affect the basal activity of the receptor, by selectively stabilizing one conformation. The T88$^{3.36}$A mutant is a CIM that reduces the basal activity [19], by selective thermal stabilization of the antagonist-bound conformation [20]; however no significant effect was observed on the basal activity for the S277$^{7.42}$A mutant [19]. On the other hand, the same mutation can directly modulate the binding affinity of the (partial) agonist for the effective, active conformation. While there is no experimental data for the conformational affinity, our approach allows to model each of these effects independently and combine them a posteriori, taking advantage of the possibility of combining two thermodynamic cycles if they share a common leg.

As shown in Fig 1C, the effect of a mutation on the predicted basal activity can be modeled by designing a thermodynamic cycle that represents the effect of the corresponding Ala mutation on the conformational selectivity. The cycle can be solved through FEP simulations of the vertical legs, i.e. annihilation of the sidechain of interest (wt → Ala perturbation) in each receptor state ($\Delta G_{m,R}$ and $\Delta G_{m,R^*}$ for the inactive and active state, respectively) as:

$$\Delta\Delta G_{R\rightarrow R*} = \Delta G_{cs,wt} - \Delta G_{cs,mut} = \Delta G_{m,R*} - \Delta G_{m,R}$$

Such model assumes that the mutational effects on the GPCR *basal activity* are indeed due to differences on the conformational selection of the receptor ($\Delta G_{cs}$, for wt or mut versions of the receptor). This model can be combined with a second thermodynamic cycle, accounting for the mutational shifts in ligand binding *affinity* for the *active* conformation, which we extensively used to explain mutagenesis effects on A$_{2A}$AR agonist binding [34]. The addition of these two cycles (Fig 5A) should yield as a net result the effect the mutation on the ligand internal *efficacy*, estimated as the difference between the two vertical legs delimiting the combined thermodynamic cycle in Fig 5A, i.e. $\Delta\Delta G_{EC_{50}}^{calc} = \Delta G_{R*+L} - \Delta G_R$ (Fig 5B, solid blue columns). It should be noted that, in this case, the experimental parameter that we are trying to match with these relative free energy calculations is not the maximum efficacy (%E$_{max}$), but the internal efficacy of each compound (EC$_{50}$), which can vary upon receptor mutations. Thus, a numerical (quantitative) correlation between calculated and experimental values can be attempted in this case, by expressing the experimental shift in EC$_{50}$ values induced by a mutation as $\Delta\Delta G_{EC_{50}}^{exp} = RTln^{EC_{50(wt)}}/_{EC_{50(mut)}}$ (Fig 5B, solid orange columns). Finally, the additional calculation of the shared vertical leg, $\Delta G_{R^*}$, which is canceled in the combined cycle, allows to separate the effect on basal activity ($\Delta\Delta G_{R^*}$, Fig 5A, left side, Fig 5B, dense dashed columns) and on ligand affinity for R$^*$ ($\Delta\Delta G_{L-R^*}$, Fig 5A, right side; Fig 5B, light dashed columns). As we will see, this can provide valuable information for the interpretation of the calculated data.

We first looked into the effects on the basal activity of the receptor of the T88$^{3.36}$A and S277$^{7.42}$A mutations. According to our model, the CIM T88$^{3.36}$A should increase the relative stability of the inactive receptor, which is precisely the outcome of the corresponding FEP simulations (Fig 5B). The active state of the T88$^{3.36}$A mutant is significantly less stable than the wt, with $\Delta\Delta G_{R^* \rightarrow R}$ = 3.98 kcal/mol (S1 Table). In contrast, a similar analysis of the S277$^{7.42}$A mutation shows a negligible value for the calculated values of $\Delta\Delta G_{R^* \rightarrow R}$, (Fig 5B and S1 Table), meaning that the conformational equilibrium should not be affected by the mutation, in line with the experimentally observed neutral effect of this mutation on the basal activity of the receptor [19].

The overall mutational effects on the modulation of ligand internal efficacies were then calculated and compared to the experimental efficacy shifts in each case (Fig 5B, solid columns). One can observe that the modeled T88$^{3.36}$A mutation does not significantly affect the predicted internal efficacy of LUF5834 ($\Delta\Delta G_{EC_{50}}^{calc} = -0.23\ kcal/mol$, Fig 5A and S2 Table) in line with

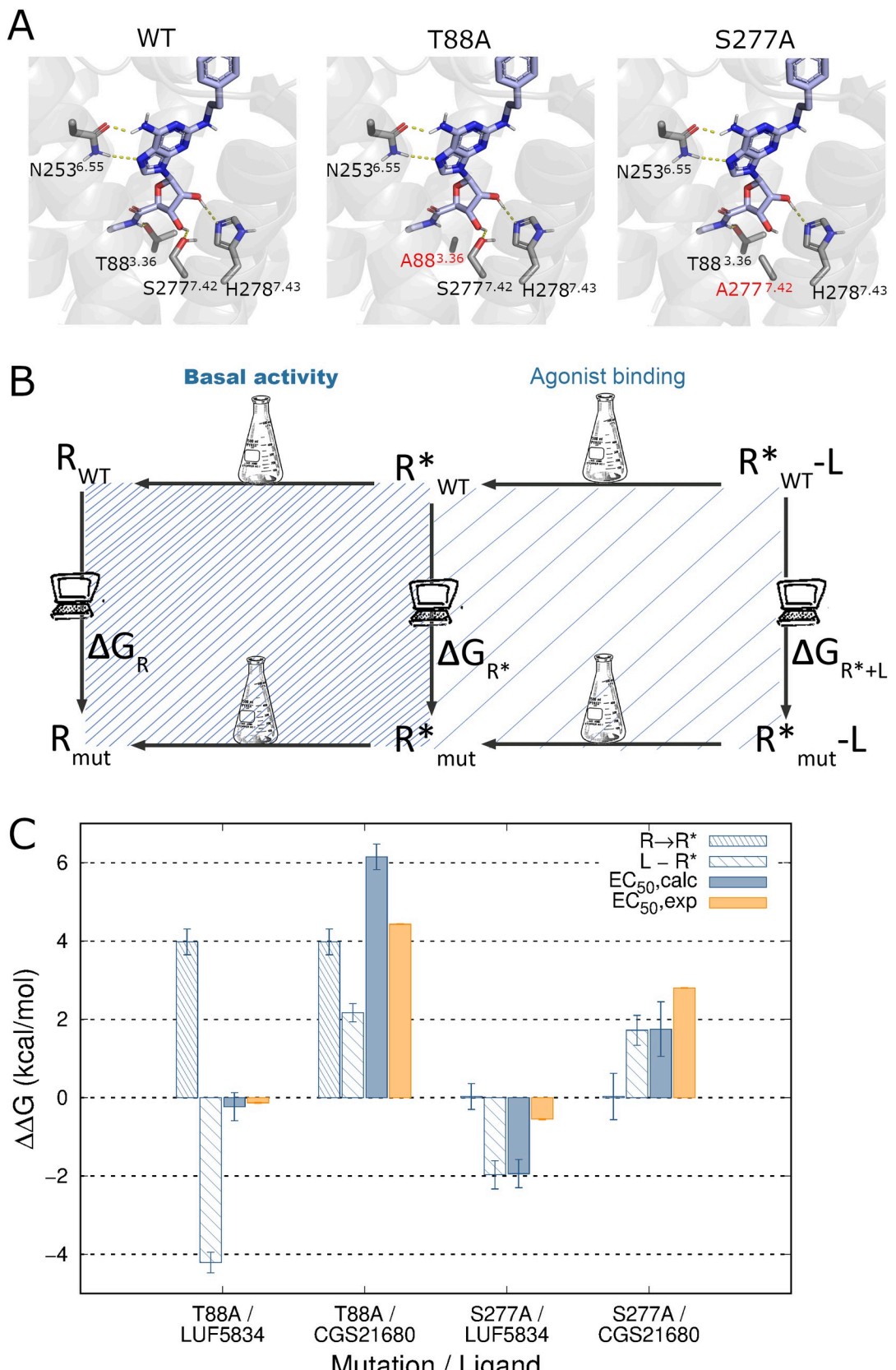

**Fig 5. Effect of protein mutations on ligand efficacy.** (A) Binding mode of the agonist CGS21680 to the WT $A_{2A}AR$ (left), T88A (center) and S277A (right) mutant forms of this receptor (mutations indicated in red), simulated by FEP simulations (H-bonds in yellow dashed lines with residues involved in sticks). (B) Thermodynamic cycles representing the effect of a point mutation (mut) on the distribution of inactive (R) and active (R*) states of the receptor (left side, dense dashed), and on the affinity of a ligand (L) for the active state (right side, light dashed). The combination of the two thermodynamic cycles would yield the net effect of the mutation ligand efficacy. (C) Calculated effects of point mutations on the $A_{2A}AR$ constitutive activity ($\Delta\Delta G_{R^* \to R}$, dense dashed columns), and on the ligand relative affinity for R* ($\Delta\Delta G_{L-R^*}$, light dashed columns), following the corresponding thermodynamic cycles depicted in (A). The overall effect of the mutation on the shift in ligand efficacy, ($\Delta\Delta G_{EC_{50}}^{calc}$, solid blue) is calculated by combination of these two values, showing correlation with the experimental values ($\Delta\Delta G_{EC_{50}}^{exp}$, solid orange, see text).

the experimental data ($\Delta\Delta G_{EC_{50}}^{exp} = -0.13$ $kcal/mol$) [19]. A deeper look at the data allows us to envisage a mechanism for this neutral effect, since the relative increase in affinity for the active state is compensated by a decrease in the population of this active state induced by the mutation (light and dense dashed bars, respectively, in Fig 5B). Conversely, the model indicates that the S277$^{7.42}$A mutation does not affect the conformational equilibrium of the receptor, and in this case the predicted increase in affinity of LUF5834 for the active state is translated into a net increase of the efficacy of this ligand upon mutation ($\Delta\Delta G_{EC_{50}}^{calc} = -1.94$ $kcal/mol$), in qualitative agreement with the experiment data ($\Delta\Delta G_{EC_{50}}^{exp} = -0.55$ $kcal/mol$,) [19]. For the full agonist CGS21680, both mutations resulted in a similar decrease in estimated ligand affinities, in agreement with our previous computational modeling of the mutagenesis data for ribose-containing ligands [34]. However, this effect is amplified by the diminished availability of the active state upon T88$^{3.36}$A mutation, resulting in a much more drastic estimated reduction of CGS21680 efficacy ($\Delta\Delta G_{EC_{50}}^{calc} = 6.15$ $kcal/mol$) as compared to the S277$^{7.42}$A mutant ($\Delta\Delta G_{EC_{50}}^{calc} = 1.75$ $kcal/mol$). These results allow a correct ranking of this pair of mutations in terms of how they affect the internal efficacy of this ligand, in line with the experimental observation where the reduction in the EC50 of CGS21680 was 10 times higher for the T88$^{3.36}$A mutant [19].

## Discussion

The use of FEP simulations coupled to a combination of thermodynamic cycles to study the effect of mutations on binding and catalysis of subtilisin, was first presented in a seminal paper of Rao *et al.* in the late eighties [35]. Starting from that idea, in this work we introduce a novel approach to estimate the modulation of GPCR activation, based on an original design of thermodynamic cycles connecting receptor conformations. These thermodynamic cycles (Figs 1 and 5) resemble the pharmacological representation of GPCR activation, which are the basis for the estimation of the corresponding equilibrium constants [10]. In our case, we compare the effect on the activation pathway between chemically related species, these being either pairs of ligands or single-point mutants *vs* the wt receptor. The combination of this approach with the use of spherical boundary conditions around the binding results in very efficient computational modeling of the modulation of GPCR activation. Indeed, with such reductionist model one can indistinctively use any active-like structure of the receptor (i.e., with or without the bound G-protein, the last being our choice), since the RMSD of the binding site is typically very low (in the case of $A_{2A}AR$, 0.69 Å for the Cα atoms). It is worth mentioning that this model is not intended to model the activation pathway *per se*, but instead the variations in the activity of the receptor induced by different types of modulators (i.e., external ligands, mutations).

Taking additional advantage of the increased structural information of GPCRs in different conformations, as is the case of the $A_{2A}AR$, one can solve the vertical legs of the designed cycles via automated FEP protocols tailored for ligand [25] or residue [26] perturbations,

respectively. Following the scheme depicted in Fig 1B, we demonstrate the validity of this approach with the calculation of structure-efficacy relationships for three series of $A_{2A}AR$ ligands, resulting in significant discrimination of full and partial agonists from neutral antagonists (Figs 3 and 4). Indeed, the qualitative correlation between the calculated $\Delta\Delta G$ and the $\Delta$-efficacy is achieved in up to 70% of the pair comparisons performed, including those where the agonist/antagonist profile differences are more significant (i.e. NECA, ADO as full agonists, or LUF5835 and compound **10n** as the most potent partial agonists within their chemotype). The agonistic profile is modeled as the capacity of a molecule to achieve the desired conformational selectivity for the active configuration, opening the door to the structure-based computational design of compounds with tailored pharmacology, with additional potential to provide insights in pharmacogenomics of drugs [36].

Using the analogous approach depicted in Fig 1C, we show how to estimate the effect of protein mutations on the conformational equilibrium of the receptor, which can be translated to variations on its basal activity. The approach is applied to characterize a constitutively inactive mutation (CIM, Thr88$^{3.36}$A) as well as a neutral mutation (Ser277$^{7.43}$A), finding the same discrimination as the experimental data available in both cases (Fig 5B). Moreover, taking advantage of the additive property of the thermodynamic cycles, one can model ligand efficacy as the estimation of the ligand affinity for the active (R$^*$) conformation weighted by the accessibility of the ligand to R$^*$. The combined cycle, shown in Fig 5A, allowed a successful discrimination of the different effects exerted by each of these two mutations on the efficacy of a full and a partial agonist. The relevance of this result goes beyond the successful correlation of the computed ranking of the mutational shifts in ligand efficacy with the experimental values, and additionally provides a structural and mechanistic framework to interpret these results. Thus, it was found that the CIM T88$^{3.36}$A, by displacing the equilibrium towards the inactive conformation, can neutralize the predicted increase in affinity for the partial agonist LUF5834 or, conversely, display a synergistic effect on the predicted decrease in the R$^*$ affinity for the full agonist CGS21680, explaining the dramatic decrease on its efficacy for the $A_{2A}AR$ (Fig 5B). The mutation S277$^{7.43}$A was found to be neutral with regards to the receptor conformational equilibrium, in agreement with its negligible effect on the basal activity of the receptor. Consequently, the correctly predicted net effect in increasing or decreasing the ligand efficacy of the partial and full agonist, respectively, is directly correlated to the predicted effects of the mutation on the affinity for R$^*$ in each case (Fig 5B).

The presented approach provides a generally applicable framework, as long as the underlying pharmacological problem can be based on clear structural endpoints (in the present study, an active and inactive structure of the receptor). The rapidly expanding arsenal of GPCR structures [37], with various states available for a number of receptors [16, 38], will widen the applicability of our workflow to cover a large extent of the GPCR-ome. These structures not only include G protein ternary complexes, but lately also e.g. β-arrestin bound structures [39], or nanobody derived structures of intracellular binding partners [40]. As such, the method is not limited to the prediction of agonist profiles of compounds, but could potentially be extended to the design of biased ligands [41]. Moreover, there is now strong evidence from HDX [42] or NMR [43] experiments of intermediate conformational states of GPCRs, including recent proposals that the partial agonists of the $A_{2A}AR$ could preferentially bind one of these intermediate states with compromised G-protein coupling [43,44]. One could expect that the precise structures of those states could be structurally revealed in the near future, and consequentially be used to build more precise thermodynamic cycles representing revised pharmacological models of receptor activation [43,45]. Alternatively, the derivation of activation states from inactive receptors via MD approaches can provide structural insights into receptor activation mechanisms [46], and identify intermediate states along the receptor activation path. In

addition, the experimental structures available provide excellent starting points for the generation of reliable homology models [47], which can be also useful starting structures for FEP simulations, as we recently showed in the case of the orphan GPR139 [48], the neuropeptide Y receptor family [26,49] or the $A_{2B}AR$ [50].

To the best of our knowledge, this is the first time that the regulation of the activation of a GPCR is modeled with the use of thermodynamic cycles coupled to FEP simulations. The generated framework is easily extensible to other GPCRs, offering a computational approach to design ligands with tailored pharmacological properties or to predict the effect of point mutations on the receptor conformational equilibrium. The last question is gaining interest as we see increasing examples of GPCR point mutations related to disease, as is the case of the $A_{2B}AR$ in cancer [51], by the mechanism of shifting the basal activity to either constitutively inactive or constitutively active mutations. An additional advantage of our end-point approach, besides the computational efficiency, is its modularity, allowing the combination of thermodynamic cycles to predict e.g. shifts in ligand efficacy induced by point mutations. Moreover, the protocol could potentially be extended to, e.g., DFG in and out kinases [52], open and closed state of ion channels [53] or single solute carrier (SLC) transporters [54], provided that the chemical space of the studied cases sufficiently overlap (i.e. point mutations or related chemotypes).

## Methods

### Structure preparation, membrane insertion and equilibration

The high-resolution crystal structure of the $A_2AR$ with antagonist ZM241385 (PDB code 4EIY [12]) was used as receptor starting structure for the inactive state, whereas the same receptor in complex with agonist NECA (PDB code 2YDV [31]) was used for the active (like) state. In the first case, the engineered BRIL fusion protein was removed and missing loops (C-terminal fragment of EL2 and most of EL3) were modelled, and protonation states of residues assigned, as described elsewhere [55]. Notably, this included the allosteric sodium ion and the charged form of the coordinating residue Asp50$^{2.50}$. The active-like structure, on the other side, cannot accommodate this sodium ion though the protonation state of Asp50$^{2.50}$ remains charged [55]. The structure used here (2YDV) contained four thermostabilizing mutations L48A$^{2.46}$, A54L$^{2.52}$, T65A$^{2.63}$ and Q89A$^{3.37}$, which we reverted prior to a similar assignment of protonation states as described previously [56], and finally it was aligned to the inactive structure of the receptor. The selection of this simpler structure to represent R$^*$ in our models over the fully active G-protein bound $A_{2A}AR$ (available with PDB code 5G53), was based on the fact that the RMSD between both binding sites (i.e. considering the spheres to simulate for FEP calculations, see next section) was as low as 0.69 Å for the Cα trace. In addition, it proved previously a very useful framework to reproduce, with high correlation, the effect of point mutations to agonist binding on this same structure [34, 56]. The 3D coordinates for the ligands were generated with LigPrep and subsequently docked to the prepared $A_{2A}AR$ structure using Glide [57], and the antagonists structures by flexible ligand alignment to their reference agonist compound. Apo structures were generated by removing the ligand, but keeping crystallographic waters, and subsequently embedded in a solvated membrane environment using PyMemDyn [58]. This protocol embeds a structure in a pre-equilibrated POPC (1- palmitoyl-2-oleoyl phosphatidylcholine) membrane model such that the TM bundle is parallel to the vertical axis of the membrane. The system is then soaked with bulk water and inserted into a hexagonal prism-shaped box that is energy-minimized and carefully equilibrated during 5 ns, following the PyMemDyn protocol described elsewhere [58]. The standard OPLS all-atom (OPLS-AA) force field is used for all residues [59], and parameters for membrane lipids were taken from the Berger united-atom model

[60]. The corresponding equilibrated holo structures were generated by restoring the docked ligands and removing overlapping water molecules.

## FEP simulations

The receptor-ligand structures in the equilibrated membrane were subsequently transferred to the MD package Q (version used available at https://github.com/esguerra/q6), in order to perform FEP calculations under spherical boundary conditions [61]. A 50 Å diameter sphere was centered on the center of geometry of ZM241385 (or equivalent point in the remaining structures), where solvent atoms are subject to polarization and radial restrains using the surface constrained all-atom solvent (SCAAS) model to mimic the properties of bulk water at the sphere surface [62]. Atoms lying outside the simulation sphere are tightly constrained (200 kcal/mol/Å$^2$ force constant) and excluded from the calculation of non-bonded interactions. Within the simulation sphere, long range electrostatics interactions beyond a 10 Å cut off were treated with the local reaction field method [63], except for the atoms undergoing the FEP transformation, where no cutoff was applied. Solvent bond and angles were constrained using the SHAKE algorithm [64]. All ionizable residues outside the sphere and those within the boundary were considered in their neutral form as described elsewhere [25]. Residue parameters were translated from the latest version of the OPLSAA/M force field [59], whereas ligand OPLS2005 parameters were retrieved from Schrodinger's ffld_server [65], and translated to Q following the QligFEP protocol [25]. The simulation sphere was heated from 0.1 to 298 K during a first equilibration period of 0.61 nanoseconds, where the initial restraint of 25 kcal/mol·Å$^2$ imposed on all heavy atoms was slowly released. Thereafter the system was subject to unrestrained MD simulations, starting with a 0.25 nanosecond unbiased equilibration period which is followed by the FEP sampling, applying different protocols for sidechain and ligand perturbations as detailed below. In both cases, atom transformations occur between initial and ending states, evenly divided into a number of λ windows that depend on the FEP protocol adopted (see below). This sampling is replicated in 10 independent MD simulations with different initial velocities, each of them consisting of 10 ps sampling per λ window using a 1 fs time step in all cases.

The generalized FEP protocol for amino acid mutations, QresFEP [26,66] was used to estimate the effects of single point mutations on ligand binding (available at https://github.com/qusers/qligfep). Briefly, QresFEP is a single-topology FEP protocol that divides the sidechain perturbation to alanine into separate stages, where atom annihilations occur gradually for each charge group (as defined on the OPLS force field), starting from the most topologically distant from the C$_\beta$ atom, in four consecutive stages [66]: 1) the partial charges are initially removed, 2) van der Waals potentials are transformed into smoother soft-core potentials, 3) annihilation of the corresponding group of atoms., 4) restoring the partial charges of the final species. The number of perturbation stages needed for the full annihilation depends on the nature of the sidechain involved, in this case ranging from four (Ser) to five (Thr), where each of the subsequent stages is evenly divided into 20 λ windows. To fulfill a thermodynamic cycle, the same sidechain annihilation is simulated in the apo structure of the protein, so that the energetic difference between these two processes equals the binding affinity shift due to the mutation. It follows that the sampling time for the sidechain perturbation here considered was 8–10 ns. These sampling times have been repeatedly shown to be sufficient for conformational sampling at equilibrium [26,66], assuming that only minor local conformational changes occur as is the case of the models here presented.

All ligand perturbations were performed with our dual-topology QligFEP protocol [25], where the full transformation of one ligand into another is performed along a linear λ

sampling consisting of 50 windows (code available at https://github.com/qusers/qligfep). In order to fulfill the thermodynamic cycle, the simulation is performed in the two receptor states to compare, i.e. active (R*) and inactive (R), and in this case the difference between the ligand transformation in the two states equals the difference in relative binding affinity between the two ligands.

In both residue and ligand transformations, the sampling time was considered sufficient as judged from the low s.e.m. estimated over replica simulations (see Results). Relative binding free energies were estimated by solving the thermodynamic cycle utilizing the Bennet acceptance ratio method (BAR) [67] as

$$\Delta G_i = -\beta^{-1} ln \frac{\langle 1 + e^{-\beta(\Delta U \Delta \lambda_i - C_i)} \rangle_{i+1}}{\langle 1 + e^{+\beta(\Delta U \Delta \lambda_i - C_i)} \rangle_i} + C_i \tag{1}$$

where the constants $C_i$ are optimized iteratively so that the two ensemble averages become equal, yielding $\Delta G_i = C_i$. Average values ± SEM are reported from the 10 independent MD simulations in each relevant state.

Starting structures, input files and submission scripts needed to reproduce the simulations, together with raw output data, are available at https://zenodo.org/record/5602896#.YXlGHZ5ByF4.

## Supporting information

**S1 Table. Calculated and experimental relative binding free energies (kcal/mol) of the pairs of agonist/antagonist compounds of the A2AAR (data corresponding to Fig 4 in the main text).**
(PDF)

**S2 Table. Calculated effects of point mutations on the constitutive activity (expressed as the shift in the distribution between R and R*, $\Delta\Delta$GR* →R), and on the relative affinity of each ligand for the active receptor ($\Delta\Delta$Gbind, R*).** The effect of the mutation on the change in efficacy is calculated by adding these two values, according to the thermodynamic cycle shown in Fig 5 ($\Delta\Delta G_{EC50}$).
(PDF)

## Acknowledgments

Computations were performed on resources provided by the Swedish National Infrastructure for Computing (SNIC). Additional support from the Swedish strategic research programme eSSENCE is acknowledged. The authors participate in the European COST Action CA18133 (ERNEST).

## Author Contributions

**Conceptualization:** Willem Jespers, Adriaan P. IJzerman, Hugo Gutiérrez-de-Terán.

**Formal analysis:** Willem Jespers, Laura H. Heitman, Adriaan P. IJzerman, Eddy Sotelo, Johan Åqvist, Hugo Gutiérrez-de-Terán.

**Investigation:** Willem Jespers, Hugo Gutiérrez-de-Terán.

**Methodology:** Willem Jespers, Johan Åqvist, Hugo Gutiérrez-de-Terán.

**Project administration:** Johan Åqvist, Hugo Gutiérrez-de-Terán.

**Resources:** Hugo Gutiérrez-de-Terán.

**Software:** Willem Jespers, Johan Åqvist, Hugo Gutiérrez-de-Terán.

**Supervision:** Johan Åqvist.

**Validation:** Laura H. Heitman, Eddy Sotelo.

**Writing – original draft:** Willem Jespers, Hugo Gutiérrez-de-Terán.

**Writing – review & editing:** Willem Jespers, Laura H. Heitman, Adriaan P. IJzerman, Eddy Sotelo, Gerard J. P. van Westen, Johan Åqvist, Hugo Gutiérrez-de-Terán.

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
