## [Decision Letter · Decision Letter 0]

9 Jul 2021

Dear Dr. Gutierrez de Teran:

Thank you very much for submitting your manuscript "Deciphering conformational selectivity in the A2A adenosine G protein-coupled receptor by Free Energy simulations" (PCOMPBIOL-D-21-00923) for review by PLOS Computational Biology. 

As with all papers reviewed by the journal, your manuscript was reviewed by members of the editorial board and by several independent reviewers. Based on the reviews, we regret that we will not be pursuing this manuscript for publication at PLOS Computational Biology.

The reviews are attached below this email, and we hope you will find them helpful if you decide to revise the manuscript for submission elsewhere. 

While we cannot consider your manuscript further for publication in PLOS Computational Biology, we would like to offer you the option to transfer your submission, with reviews, to PLOS ONE https://www.editorialmanager.com/PONE/

If you DO wish to transfer your submission, please click this link:

<DeepLinkData><DeepLinkTypeID>27</DeepLinkTypeID><peopleID>592113</peopleID><userSecurityID>ee7d8438-9e77-43d5-b9bd-533de1e739bc</userSecurityID><documentID>29844</documentID><revision>0</revision><manuscriptNumber>PCOMPBIOL-D-21-00923</manuscriptNumber><docSecurityID>e4e0581d-bf03-4add-babd-d49abdf70cfb</docSecurityID></DeepLinkData>

If you do NOT wish to transfer your submission, please click this link to decline:

<DeepLinkData><DeepLinkTypeID>28</DeepLinkTypeID><peopleID>592113</peopleID><userSecurityID>ee7d8438-9e77-43d5-b9bd-533de1e739bc</userSecurityID><documentID>29844</documentID><revision>0</revision><manuscriptNumber>PCOMPBIOL-D-21-00923</manuscriptNumber><docSecurityID>e4e0581d-bf03-4add-babd-d49abdf70cfb</docSecurityID></DeepLinkData>

Please note, all PLOS journals are editorially independent and vary in submission requirements.

Should you choose to transfer, your manuscript files, along with the reviewers' comments and their identities will be transferred automatically, and you will receive a confirmation email within 24 hours. Once transferred, your submission will be returned to you so you can check over your record before completing the submission. You may be asked to provide additional information, such as a response to the reviewers' comments. If you have any questions, please contact the editorial office of PLOS ONE https://www.editorialmanager.com/PONE/

We are sorry that the news is not more positive on this occasion, and we hope you will consider PLOS Computational Biology for future submissions. Thank you for your support of PLOS and of open-access publishing.

Sincerely,

Michael Gilson

Guest Editor

PLOS Computational Biology

Nir Ben-Tal

Deputy Editor

PLOS Computational Biology

Reviewer's Responses to Questions

**Comments to the Authors: **

Reviewer #1: Jespers et al present a study in which FEP calculations are used to probe the molecular basis of different degrees of agonism in small-molecule ligands and to explain the effects of mutations at the A2A adenosine receptor. 

Major comments

1. It's not clear from the manuscript how the two distinct "active" structures mentioned in the methods are used in the calculation: 

"The high-resolution crystal structure of the A2AR with antagonist ZM241385 [4EIY (12)] was used as receptor starting structure for the inactive state, whereas 2YDV (31) and 5G53 (14) were used for the active state. ... The active structure ..."

I find this confusing, especially since it is mentioned elsewhere in the manuscript that only an active and inactive structure is needed. Was one or two "active" structures used? Moreover, 2YDV is not really fully active, but has been termed intermediate. How is the intermediate nature of this conformation taken into account? In addition, 2YDV has four thermostabilizing mutations in the receptor (L48A^2.46, A54L^2.52, T65A^2.63 and Q89A^3.37) that are absent in the receptor in 5G53 (I believe these are also absent in 4EIY). How are these mutations taken into account, as they might affect the results? Why not use the 3QAK structure (assuming an intermediate/active-like conformation is desired), which I believe is wild type?

2. When discussing molecular mechanisms of mutations/ligands, it would be extremely helpful to have clear renderings that visually communicate these ideas. Currently, there is only one figure in the manuscript that shows 3D structures (of binding modes, in this case). I fear that, for anyone who is not already familiar with this system, it will be difficult to envision how these mutations/chemical modifications work by just looking at figure 3 and the various bar plots.

Additional comments:

1. It would be helpful to be more explicit when describing the system preparation. For instance, was sodium present in the inactive simulations? Why/why not? How might this affect the results? Similarly, D2.50 is most likely charged (and sodium-bound) in the inactive state, but can be protonated (neutral) in the active state (I believe this is the case for, e.g., B2AR). What was the protonation state of D2.50, and how might the protonation state of this residue affect the results?

2. Is 10 ps sampling per λ window really sufficient?

Reviewer #2: In this manuscript, the authors present a model of GPCR activation based on thermodynamic cycles connecting receptor states (e.g. antagonist-/agonist-bound, wild type/mutant) and conformations (inactive/active). This allows a ‘pharmacological representation’ of GPCR activation and, in principle, to separate the effects of ligand and receptor on activation. The model also provides a structural and mechanistic framework to interpret data on the A2A adenosine receptor, used as a model system. The ΔG values corresponding to certain branches of the thermodynamic cycles are calculated using free-energy-perturbation molecular dynamics simulations. The values for other branches are estimated from published pharmacology and activity data.

The first part of the results –study of structure-efficacy relationships of a series of ligands– is hard to follow for a non-specialist in the pharmacology of A2A adenosine receptors. There are also small aspects that do not help the reader. For instance, Figure 4 is not very informative. One cannot easily judge the relevance of the similarities and differences in the bars, as the labels in the X axis do not mean much and one needs to constantly refer to Figure 2 to understand the significance of the data. Overall, according to the way this part of the manuscript is written, it seems too specialized and more suited to a journal on medicinal chemistry. The second part –study of the effect of point mutations on basal activity and conformational selectivity– is better explained and more clear, and conveys better the possibilities of this methodology.

It seems that only Figure 4 and 5A contain new data (two bar plots) –plus one supplementary table. This makes the manuscript feel ‘light’ on analysis, which is not compensated by the current text. This work presents an interesting way to use thermodynamic cycles and molecular simulations to study structure-activity relationships. However, the manuscript in its present form seems too specialized and more suited to a molecular pharmacology journal.

# Minor

- I don’t see the point of Figure 1A. It only says that there are two states of the receptor.

- Grammar and syntax should be revised once more.

- The authors might consider finding more descriptive names for the ligands that do not require checking constantly Figure 2.

- The thermodynamic cycles (Figures 1BC and 5A) would be much more clear if they displayed the corresponding ΔG in the branches instead of cartoons. It would also help if they included the final formula for ΔΔG, for instance.

Reviewer #3: The authors have presented an original implementation of a very common technique in free energy perturbation methods, namely the thermodynamic cycle. There are a number of interesting things claimed to come out of this approach, as exemplified in Figures 4 and 5.

1. The effect of a mutation on basal activity (R->R* in Fig 5)

2. Modelling ligand efficacy as a combination of ligand affinity for the active state R* weighted by the accessibility to R*

3. Modelling the activation of a GPCR using FEP/a thermodynamic cycle 

4. Predict the pharmacological profile of a series of full and partial agonists / antagonists

p21/p17 is key to the article as it goes through the main calculations.

On balance I am very much in favour of publication and I hope the authors will be able to make significant improvements to the clarity of their article.

I find that 1 is convincing - this is very useful data and could find widespread application. 1. contributes to 2, and it is nice to see how this decomposition into different effects contributes to the overall picture, giving deeper molecular insight. However, only two examples are given (T88A and S277A), and so the authors have a high probability of getting the correct result by chance. It would be more convincing if there were 2 more examples, with one extra mutation that changes basal activity and 1 extra mutation that does not.

3. This is a bold claim, but not too much is made of this in the manuscript as the effect is indirect.

4. I do not find point 4 (Fig. 4) convincing as some of the bars in Figure 4 are very small - so I am not convinced that there is real discrimination between between say 10a and 10b. It maybe that the data needs to be described more carefully.

Minor Comments

The labelling in Fig. 2 is confusing as 10a is an antagonist and 10b is a partial agonist - but the panel shows the structure of a and b to be the same. There is probably a better way to do this: inclusion of a dotted line as for LUF may help.

p20/16 T288(3.36)... antagonist -bound conformation,..... What follows is a new theme regarding S277(7.42) and so the comma should be a semicolon (;).

p20/16 ligand internal efficacy is an interesting concept, but it is new and so difficult to get your head around it. I think it would be clearer if the corresponding equation were put in parentheses after every time the phrase is used (as it is not used very often) to help with reader comprehension. We also have predicted internal efficacy (p21/17) and mutational ligand efficacy (Fig. 5). The various incarnations of internal efficacy are not helpful but rather add to confusion - are they the same thing or not?

p22/18 we have thin and thick dashed bars - but we have dense bars in Fig. 5. The descriptions need to the consistent. Dense is probably better as we don't know whether the bars are blue or white so think and thick are interchangeable.

p22/p18. The increases and decreases can be a little difficult to follow as there is a delta delta so the sign convention gets somewhat diluted. Where it says a predicted increase in affinity of LUF5834 (~ -0.2 kcal/mol)...

I suggest adding the numerical value to the text would make it easier to confirm that we are looking at the right bars.

Somewhere, the authors should note that R* is an oversimplification

p26/22 There is a reference to intermediate states - the authors should elaborate how these would be identified in such a framework - would it require an X-ray structure of an intermediate state?

p28/24. wrong use of shortly. Delete 'shortly,' and start 'This protocol...'

p30/p26 8-10 ns is very short and requires some justification. It basically assumes no conformational changes, not even minor ones?

**Have the authors made all data and (if applicable) computational code underlying the findings in their manuscript fully available?**

Reviewer #1: No: The authors could share the data used to make the various plots, the analysis code, as well as simulation trajectories in a public repository.

Reviewer #2: No: Most of the data is only presented as bar plots in Figure 4 and 5A.

Reviewer #3: No: The availability or otherwise of the in-house FEP software should be made clear

PLOS authors have the option to publish the peer review history of their article (what does this mean?). If published, this will include your full peer review and any attached files.

Reviewer #1: No

Reviewer #2: No

Reviewer #3: No

---

## [Decision Letter · Decision Letter 1]

21 Oct 2021

Dear Dr. Gutierrez de Teran,

Thank you very much for submitting your manuscript "Deciphering conformational selectivity in the A2A adenosine G protein-coupled receptor by free energy simulations" for consideration at PLOS Computational Biology. As with all papers reviewed by the journal, your manuscript was reviewed by members of the editorial board and by several independent reviewers. The reviewers appreciated the attention to an important topic. Based on the reviews, we are likely to accept this manuscript for publication, providing that you modify the manuscript according to the review recommendations, including the provision of "trajectories and analysis code in a public repository such as Zenodo", as specified by Reviewer 1.

Sincerely,

Michael Gilson

Guest Editor

PLOS Computational Biology

Nir Ben-Tal

Deputy Editor

PLOS Computational Biology

[LINK]

Reviewer's Responses to Questions

**Comments to the Authors:**

Reviewer #1: The manuscript is improved, and the authors have addressed many of my concerns.

However, regarding my comment on the structures used in the study, I think the authors should clearly state (in the manuscript) that they used an agonist-bound, but not G protein–bound, structure (which the authors themselves refer to as "pseudo-active" in the introduction) to represent the active state. I think this is all the more important because Fig. 1 shows a G protein–bound active receptor. I think the reader will assume that the active structure shown in Fig. 1 was used for the simulations (the structures used for the rendering should also be specified in the caption). Since a G protein–bound structure is available for this receptor, it would be important to briefly explain why that structure was not used as the active state.

Reviewer #3: I am pleased that the authors have been permitted to resubmit to PLOS computational biology as this manuscript lies better here than in PLOS one.

This is a vastly improved manuscript, which I think should be accepted. I just have a minor comment on the legend of Fig. 1, which can be clearer. e.g. what is meant by the neutral effect of an antagonist/ Would delta G really be zero? I know what is intended, but it probably needs wording differently

**Have the authors made all data and (if applicable) computational code underlying the findings in their manuscript fully available?**

Reviewer #1: **No: **Trajectories and analysis code should be made available in a public repository such as Zenodo

Reviewer #3: Yes

PLOS authors have the option to publish the peer review history of their article (what does this mean?). If published, this will include your full peer review and any attached files.

Reviewer #1: No

Reviewer #3: No

Figure Files:

Data Requirements:

Reproducibility:

References:

---

## [Editor Report · Decision Letter 2]

11 Nov 2021

Dear Dr. Gutierrez de Teran,

We are pleased to inform you that your manuscript 'Deciphering conformational selectivity in the A2A adenosine G protein-coupled receptor by free energy simulations' has been provisionally accepted for publication in PLOS Computational Biology.

Best regards,

Michael Gilson

Guest Editor

PLOS Computational Biology

Nir Ben-Tal

Deputy Editor

PLOS Computational Biology

---

## [Editor Report · Acceptance letter]

18 Nov 2021

PCOMPBIOL-D-21-00923R2 

Deciphering conformational selectivity in the A2A adenosine G protein-coupled receptor by free energy simulations

Dear Dr Gutiérrez-de-Terán,

I am pleased to inform you that your manuscript has been formally accepted for publication in PLOS Computational Biology. Your manuscript is now with our production department and you will be notified of the publication date in due course.

With kind regards,

Katalin Szabo
